# Well Differentiated Grade 3 Neuroendocrine Tumors of the Digestive Tract: A Narrative Review

**DOI:** 10.3390/jcm9061677

**Published:** 2020-06-01

**Authors:** Anna Pellat, Romain Coriat

**Affiliations:** 1Department of Gastroenterology and digestive oncology, Cochin Teaching Hospital, AP-HP, 75014 Paris, France; romain.coriat@aphp.fr; 2Faculté de Médecine, Université de Paris, 75006 Paris, France; 3Oncology Unit, Hôpital Saint Antoine, AP-HP, Sorbonne Université, 75012 Paris, France

**Keywords:** well-differentiated, grade 3, neuroendocrine tumor

## Abstract

The 2017 World Health Organization (WHO) classification of neuroendocrine neoplasms (NEN) of the digestive tract introduced a new category of tumors named well-differentiated grade 3 neuroendocrine tumors (NET G−3). These lesions show a number of mitosis, or a Ki−67 index higher than 20% with a well-differentiated morphology, therefore separating them from neuroendocrine carcinomas (NEC) which are poorly differentiated. It has become clear that NET G−3 show differences not only in morphology but also in genotype, clinical presentation, and treatment response. The incidence of digestive NET G−3 represents about one third of NEN G−3 with main tumor sites being the pancreas, the stomach and the colon. Treatment for NET G−3 is not yet standardized because of lack of data. In a non-metastatic setting, international guidelines recommend surgical resection, regardless of tumor grading. For metastatic lesion, chemotherapy is the main treatment with similar regimen as NET G−2. Sunitinib has also shown some positive results in a small sample of patients but this needs confirmation. Peptide receptor radionuclide therapy (PRRT) and immunotherapy could be future available treatments after ongoing studies. The goal of this review was to sum up the latest data on the epidemiology and management of digestive NET G−3.

## 1. Introduction

Neuroendocrine neoplasms (NEN) of the digestive tract are rare tumors with a rising incidence due to diagnostic improvement [1]. Progress in pathological diagnosis has allowed a better identification and classification of these tumors. Because of the rising number of specimens showing well differentiated morphology and high Ki−67 index (>20%) [2,3,4], the 2017 World Health Organization (WHO) classification has been updated with the introduction of a new category named well-differentiated grade 3 neuroendocrine tumors (NET G−3) [5,6]. This was initially validated for pancreatic sites but was then expended to all NET G−3 tumors of the digestive tract.

Before 2017, all G−3 neuroendocrine lesions were considered as one entity named neuroendocrine carcinomas (NEC) and were equally evaluated in clinical studies [7]. Therefore, therapeutic data regarding NET G−3 specifically is still scarce. Digestive NEC are poorly differentiated tumors with a high Ki−67 index, with poor prognosis [7]. The main treatment in NEC is chemotherapy, with platinum-based regimen as standard first line [7,8]. It is well documented that well-differentiated NEN have a better prognosis, even for G−3 lesions [1,2,9]. Therefore, NET G−3 should benefit from a different therapeutic approach but more studies are needed to validate their precise management [10,11]. We present in this review the latest data on histopathological identification, incidence, treatment and outcome of NET G−3 of the digestive tract.

## 2. Histopathological Characteristics of Digestive Net G−3

### 2.1. Differentiation and Proliferation

NEN can occur anywhere in the body but are most commonly found in the lungs and in the digestive tract. They are defined by the expression of specific diagnostic biomarkers such as synaptophysin and chromogranin A (CGA) [6]. Chromogranin A staining may be lacking in high-grade NEN. In two studies, chromogranin A was present in 91% and 100% of NET G−3, compared with 75% and 89% of NEC [12,13].

Cell differentiation and proliferation rate (mitoses in high-power fields (HPF) and Ki−67 index) are both major prognostic markers used in the NEN WHO staging and grading system [5,6]. In 2017, it has been updated to integrate NET G−3. The proliferation rate separates NEN in: low-grade or G−1 (mitoses ≤ 2/10 HPF and Ki−67 index ≤ 2%); intermediate-grade or G−2 (mitoses 2– 20/10 HPF, or Ki−67 index 3–20%); and high-grade or G−3 (mitoses > 20/HPF, or Ki−67 index > 20%). Assessment of both Ki−67 and mitotic index is important to evaluate proliferation [14]. Taking cell differentiation into account, a distinction can be made between well-differentiated NET of the digestive tract (low-grade G−1, intermediate-grade G−2 and high-grade G−3), and poorly differentiated NEC of the digestive tract (high-grade G−3) that include small-cell and large-cell NEC (Table 1) [5]. 

For NET G−3, the Ki−67 index usually ranges from 20 to 50%, whereas NEC show higher values up to 100% [2,12]. In a study on 204 patients with gastroenteropancreatic (GEP) NEN G−3, 18% patients presented with a NET G−3 and median Ki−67 was 30%, whereas median Ki−67 was 80% for the 167 patients with NEC [12]. 

Morphological distinction between NET G−3 and NEC is not always easy in case of important tumoral heterogeneity and/or necrosis [15]. Classic descriptions of pancreatic NET cytology show plasmacytoid forms with round nuclei resembling benign islet cells. In pancreatic NET G−3, the increased proliferation is accompanied by morphologic changes such as apoptosis, mitoses, and nuclear tangles [15]. In addition, abundant cytoplasm seems more common in NET G−3 than G−2. Compared with NEC, NET G−3 have less pleomorphism and less necrosis [15]. 

Knowing that progression of a well differentiated NET to a high-grade tumor rarely occurs, it is not uncommon to perform multiple biopsies in one patient because of evolving features of the disease: this sometimes reveals the presence of a high-grade component, which is in favor of mixed grades in one patient.

The association of neuroendocrine morphology with another histological type can also result in tumor diagnosis difficulty. The other histological type must represent at least 30% of the tumor sample to talk about a mixed tumor. The 2017 WHO grading classification has integrated the notion of MiNEN (mixed neuroendocrine-non-neuroendocrine neoplasm) where any histological type can be associated, such as adenocarcinoma or others [5,6].

With all these difficulties combined, and to avoid errors in diagnosis, it is mandatory in France to have a second pathologic evaluation by a NEN expert pathologist from the TENpath group for NEC and NET G−3 of the digestive tract.

### 2.2. Molecular Biology

It has been shown recently that sporadic NET and NEC of the pancreas are genetically different. Immunohistochemical tools can be used to evaluate the genetic status of NEN, using antibodies against DAXX, ATRX, p53 and Rb1. On one hand, small-cell and large-cell digestive NEC are genetically similar with frequent inactivation of the TP53 and Rb pathways, correlated with intragenic mutations in the *TP53* and *RB1* genes. On the other hand, these genetic changes are rarely observed in well differentiated pancreatic NET [16]. Conversely, inactivating mutations in *DAXX* and *ATRX* and in *MEN1* are exclusively found in pancreatic NET [17,18]. Mutations in other components of the PI3K/mTOR signaling pathway including *PTEN*, *DEPDC5*, and *PIK3CA* have also been observed in well differentiated pancreatic NET [16,17,19]. In a case report on one patient with metastatic pancreatic NET G−3, the whole-genome sequencing of liver metastases exhibited a *TSC1*-disrupting fusion, showed a novel *CHD7–BEND2* fusion, but lacked any somatic variants in *ATRX*, *DAXX*, and *MEN1* [20]. To our knowledge, the majority of molecular biology results is focused on pancreatic NET G−3. To sum up, molecular alterations can help pathologists separate NEC from NET G−3 in addition to morphological cellular characteristics, and more data is still needed.

### 2.3. Importance of Ki−67 index

An accurate pathological assessment of the Ki−67 proliferation index appears critical in order to rigorously identify NET G−3. Technical factors such as the specimen type (biopsy or needle aspiration cytology), the staining technique or the type of antibody may potentially affect the reproducibility of Ki−67 assessment [21]. The existence of various methods of assessment, such as manual counting (MC), “eyeballing”, or digital image analysis (DIA), can also result in lack of uniformity and reproducibility. Based on the recommendations of the WHO grading system, MC of >2000 cells are the “gold standard” method used for comparison. MC and DIA seem more reliable than “eyeballing” because of marked interobserver and intra-observer variability [22,23]. This was particularly observed for the G−1/G−2 (2 to 5% range) and G−2/G−3 cutoffs (15% to >20%) [22]. Nevertheless, in another work, Ki−67 assessment by “eyeballing” was highly correlated with results in DIA [24]. Also, compared with DIA, MC and “eyeballing” tended to overestimate the Ki−67 index [22,24]. Finally, in a recent study, based on its cost/benefit ratio and reproducibility, MC on screenshot printed image appeared to be the most practical method for calculating the Ki−67 index [23].

## 3. Epidemiology and Tumor Presentation of Net G−3 

### 3.1. Incidence and Tumor Site

Despite their rarity, the incidence of NEN is rising due to better identification. High-grade NEN of the digestive tract represent a small percentage of these tumors [1,25,26]. Recently, the SEER database analysis of 162,983 patients with lung or extrapulmonary poorly differentiated NEC has shed some light on the epidemiology of these rare tumors [27]. NEC of the digestive tract are more frequently large-cell lesions, often found in the colon, esophagus, and pancreas and rarely diagnosed at a non-metastatic state [26,27]. Poorly differentiated NEC represent 7 to 21% of GEP-NEN [1,28,29,30]. 

With its recent identification and separation from NEC, precise data is scarce for NET G−3. Most of the data is retrospective and comes from reassessment and reclassification of NEC samples. Therefore, we can speculate that the NET G−3 incidence is probably underestimated.

In the prospective PRONET study of 1340 cases of NEN (lung and digestive), 778 patients presented with GEP-NEN, including 104 (13.5%) NEN G−3. From the 104 NEN G−3, the proportions of NEC, NET G−3 and mixed adeno-neuroendocrine carcinoma (MANEC) were 69% (*n* = 72), 20% (*n* = 21) and 11% (*n* = 11) respectively [31,32]. In the NORDIC study on 305 patients with GEP-NEN selected on Ki−67 > 20%, we can expect that there were some NET G−3 specimens since no pathological review was performed to evaluate differentiation [33]. Patients with pancreatic tumors had higher rates of positive somatostatin receptor imaging (SRI) (46%), lower values of Ki−67 (70% with Ki−67 < 55%) and longer overall survival, which could suggest the presence of NET G−3 in this population [33]. A 2015 study of 204 patients with GEP-NEN G−3 found 37 (18%) patients with NET G−3 and 167 (79%) with NEC [12]. In summary, it is difficult to get an idea of the true distribution between NEC and NET G−3, but with the available data it seems NET G−3 represent about one third of NEN G−3.

Regarding tumor site, studies suggest that NET G−3 are more often found in the pancreas with values ranging from 33 to 65% [3,12,31]. Other main locations are the colon/rectum and stomach, with 8 to 24% and 8 to 29% of NET G−3, respectively [12,31]. A NEC identification in these tumor sites, in addition with relatively low values of Ki−67%, should justify the re-analysis of the initial histopathological specimen to confirm diagnosis. 

### 3.2. Tumor Presentation

Well differentiated NET of the digestive tract are rather indolent neoplasms that may be associated with hormonal syndromes and sometimes with hereditary tumor syndromes such as multiple endocrine neoplasia (MEN1). On the contrary, poorly differentiated NEC are aggressive tumors very rarely associated with hormonal syndromes and with unknown risk factors [2,12,26]. Regarding age, data is contradictory with some studies suggesting that patients with NET G−3 are younger at diagnosis (median of 52 years) [12] and others finding no significant difference between NET G−2 or NEC [2,3,13,15]. Nevertheless, patients with NET G−3 are more likely to have a functional tumor compared to NEC (14–25%) [3,12]. 

One study of 12 cases of NET G−3 from various sites (lung and digestive) found increased levels of plasma CGA and Neuron specific enolase (NSE) or urinary 5HIAA in 42%, 25% and 25% of patients respectively, with no significant difference with NEC patients [3]. To our knowledge, there is no other data available on plasma biomarkers in NET G−3 of the digestive tract.

### 3.3. Functional Imaging

Regarding functional imaging, the use of fluorodesoxyglucose-PET (FDG-PET) is recommended in poorly differentiated NEC and can be a prognostic factor when positive in well differentiated NEN [34,35]. In Velayoudom-Cephise’s work, patients with NET G−3 had more positive SRI than NEC (*p* = 0.03) [3]. In Heetfeld’s study, for the 24 NET G−3 patients evaluated with SRI, 21 showed positive uptake (92% of positivity) which was significantly higher than for NEC patients [12]. Some 12 NET G−3 patients were evaluated with FDG-PET and 9 showed positive uptake (75% of positivity), which was no different from NEC [12]. As mentioned earlier, in the NORDIC study there were higher rates of SRI uptake in pancreas tumors, with the possible existence of NET G−3 specimens in this population [33]. These results indicate that NET G−3 are significantly more likely to have a positive SRI, but a positive FDG-PET does not appear to be able to distinguish NET G−3 from NEC. Further studies will be needed to evaluate if FDG-PET is more often positive in NET G−3 than NET G−2 and G−1, therefore confirming its prognostic impact.

In total, for a patient diagnosed with high-grade NEN, the presence of a secretory syndrome and positive SRI uptake should orient diagnosis towards NET G−3 rather than NEC. 

We present here the results of functional imaging for a patient with metastatic pancreatic NET G−3 (Ki−67 25%), before treatment initiation (Figure 1).

### 3.4. Prognosis

As mentioned earlier, NET G−3 of the digestive tract seem less aggressive than NEC but show a worse outcome than NET G−2. In NET G−3 groups, the overall survival was longer than in NEC: median survival of 41–99 months vs. 11–17 months [2,3,12]. In pancreatic NEN, overall survival of patients with NET G−2 and NET G−3 was similar (67.8 and 54.1 months, respectively), which was significantly higher than NEC patients (11 months) [2]. Overall, prognosis is considered as intermediate between NET G−2 and NEC, in both localized and metastatic settings [2]. 

We have summarized the results from the main studies evaluating characteristics of patients with NET G−3 (Table 2).

## 4. Treatment Options

Therapeutic management of NET G−3 of the digestive tract suffers from lack of data of well conducted clinical trials. Until recently, the majority of conducted studies had mixed NEC and NET G−3, so results were to be considered with caution. We will not describe the therapeutic management of MiNEN in this review [36].

### 4.1. Surgery and Liver-targeted Therapies

In non-metastatic NEN, European and American guidelines recommend surgical resection irrespective of tumor grading and differentiation [37,38,39,40,41,42]. For well differentiated NET, surgery is recommended without any adjuvant treatment; but it can also be performed after a neoadjuvant therapeutic approach in case of important initial tumor burden or in the metastatic setting. Based on the treatment approach for limited-stage small-cell lung cancer (SCLC), surgery can be proposed for localized NEC of the digestive tract, followed by adjuvant platinum-based chemotherapy [43,44,45,46]. Nevertheless, surgery is still debated in localized NEC because of the high risk of metastatic relapse and the absence of prospective clinical trials; so neoadjuvant chemotherapy and chemoradiation can also be considered, especially when surgical resection is difficult or at risk [37,46]. A recent study of 28 patients operated with pancreatic NEN G−3 showed that NET G−3 have similar postoperative survival compared to NEC, which was significantly lower than for NET G−2 and G−1 [47]. Both localized and metastatic tumors were evaluated in this work, so it is hard to conclude [47]. As mentioned above, we can also speculate that NEC surgical trials probably evaluated some misdiagnosed NET G−3 [33,43,44,45]. These elements also underline the difficulty of surgery indication in NET G−3. Overall, there is no specific data available for surgery in localized NET G−3 of the digestive tract, but it still appears as the first valid option irrespective of size and location.

In the metastatic setting, surgical management of NEN is controversial. The overall prognosis of these tumors is mainly based on tumor burden (especially in the liver), cell differentiation and Ki−67 index. In NET G−1 and G−2, metastatic surgery can be a valid option. Surgery of midgut NET can be performed in both localized and metastatic settings because of the risk of occlusion. Liver-directed therapies are also available treatments alone or in combination with surgery. Indeed, hepatic arterial embolization or chemoembolization (with doxorubicin or streptozotocin), as well as radioembolization, have shown good clinical, functional, and morphological responses for well differentiated NET when liver burden is important [48,49]. Due to the small size of retrospective studies, only a trend for superiority is observed for chemoembolization with a benefit that appears more important for pancreatic NET [49]. In metastatic NEC surgery is not recommended, even if some data suggest a benefit on survival [45]. For NET G−3, data is scarce but contradictory with some work suggesting similar post-surgical outcome than NEC, and others suggesting overall survival is similar to NET G−2 [2,12,47]. There is also no specific data for liver-directed therapies in NET G−3.

Overall, there is a need for well conducted surgical studies, but we can speculate that, from a surgical point of view, NET G−3 should be considered as NET G−2 in the localized setting and individually discussed in the metastatic setting. 

### 4.2. Somatostatin Analogues (SST)

The anti-proliferative effect of somatostatine analogues (SST) in GEP-NET G−1 and G−2 is based on a high expression of somatostatine receptors in these tumors and has been validated with the PROMID and CLARINET prospective trials [50,51,52]. SST, including lanreotide and octreotide long-acting-release (LAR), are mainly used for treatment of the secretory syndrome in functioning tumors and for “non-aggressive” well differentiated NET as first line treatment [41,42]. Results from the PROMID and CLARINET studies show that SST have a higher efficacy in case of low Ki−67 index, low hepatic load and slow pretreatment growth [50,51,52]. These parameters are what help define “non-aggressive” GEP-NET G−1 and G−2. In both studies there were no NET G−3 patients included. To this day there are no trials specifically evaluating the antitumoral effect of SST in this population. Therefore, the use of SST should be limited for NET G−3 and only considered with a close monitoring or for its effect on the secretory syndrome. 

### 4.3. Chemotherapy

As well as SST, chemotherapy is an important treatment in metastatic pancreatic well differentiated NET. It is commonly used as first line treatment in “aggressive” metastatic NET G−1 and G−2 or in case of disease progression under other type of treatments (SST or targeted therapy). As mentioned above, it can also be proposed in localized tumors to allow secondary surgery. 

Various combinations of first line chemotherapy have been validated in pancreatic G−1 and G−2 NET such as streptozotocin/doxorubicin [53], 5-fluorouracil/streptozotocin [53], LV5FU2/dacarbazine [54] and capecitabine/temozolomide [55]. The ECOG-ACRIN research group showed that the combination of temozolomide and capecitabine was associated with improved survival compared to capecitabine alone [56]. In view of streptozotocin’s renal toxicity and doxorubicin’s cardiac toxicity, all of these combinations can be proposed as first line treatment. In the BETTER trial, the addition of bevacizumab to a 5-fluorouracil/streptozotocin combination showed a significant disease control rate (56% of partial responses and 44% of stabilizations) [57]. Irinotecan with 5-fluorouracil (FOLFIRI) has also been considered as an option in second-line treatment of pancreatic NET, with an 80% disease control rate [58]. Finally, other regimens such as the combination of capecitabine and oxaliplatine (XELOX), or gemcitabine and oxaliplatine (GEMOX) have shown effective results in pancreatic NET [59,60]. 

In non-pancreatic metastatic NET, no randomized study has identified a standard of care [41,42]. Interferon can be proposed in cases of persistent secretory syndrome despite somatostatin analogs as it did not show any significant benefit in terms of progression-free survival versus chemotherapy (5-Fluorouracil/streptozotocin) [61,62]. Nevertheless, various combinations of chemotherapy have been evaluated for non-pancreatic NET but studies showed low response rates, especially with alkylant agents [63,64,65]. One explanation could be the strong expression of O_6_-methylguanine DNA methyltransferase (MGMT) in these tumors [66]. Cassier et al. suggested that GEMOX had some efficacy in pre-treated non-pancreatic NET G−1 and G−2 patients, with an 84% overall response rate [67]. The best results regarding chemotherapy in non-pancreatic metastatic NET were obtained with the BETTER trial which evaluated a capecitabine/bevacizumab combination [68]. This treatment was evaluated on 49 chemotherapy-naïve patients, with a response rate of 18%, and overall disease control rate of 88% and a progression-free survival of about 23 months [68]. 

In metastatic NEC (pancreatic and other sites), chemotherapy combining platinum derivatives (cisplatine or carboplatine) with etoposide is the first line treatment [8,69]. Association of irinotecan and cisplatine can also be proposed as first line chemotherapy, following the results of Nakano’s study [70]. FOLFIRI or the association of 5-fluorouracil and oxaliplatine (FOLFOX) can be administered as second line treatment in metastatic NEC [71,72]. One study has also suggested some efficacy of second line temozolomide-based regimen in digestive NEC, with 71% of response (partial response or stabilization) [73]. 

In metastatic NET G−3 the efficacy of platinum-based chemotherapy seems limited. Velayoudom-Cephise et al. reported 0% objective response in NET G−3 patients vs. 31% in large-cell NEC patients [3]. Similarly, in Heetfeld’s work, the response rates to platinum-based chemotherapy was 39% vs. 2% for NEC and NET G−3 respectively [12]. The NORDIC study on pancreatic NEN suggested a higher response to platinum-based regimen for patients with Ki−67 > 55% (42% vs. 15%), with probably a majority of NEC in this population [33]. Furthermore, a recent study of 70 patients with NEN G−3 also showed that chemotherapeutic outcome with platinum-based regimen was significantly worse in NET G−3 than NEC [13]. Finally, a Japanese study also found a 0% response rate to platinum-based chemotherapy in 21 patients with pancreatic NET G−3 [13]. All of these results suggest that chemotherapy regimen in NET G−3 should be in line with NET G−1 and G−2 rather than NEC, especially when Ki−67 < 55%. Platinum-based regimen can be proposed after individual discussion. In non-pancreatic NET G−3, no chemotherapy regimen should be considered as a standard of first-line care considering the very small amount of data available.

### 4.4. Targeted Therapies

Targeted therapies have mainly been validated for well differentiated NET. The European Neuroendocrine Tumors Society (ENETS) guidelines recommend them in second-line or in first-line when chemotherapy is not appropriate [74].

In advanced pancreatic NET G−1 and G−2, sunitinib [75], a tyrosine kinase inhibitor, and everolimus [76], an mTOR inhibitor, have both proven their efficacy in randomized phase III studies. Regarding advanced non-pancreatic NET, only everolimus has shown some efficacy. In the RADIANT 2 study everolimus with octreotide improved progression-free survival (*p* = 0.026), and this effect was confirmed by the RADIANT 4 trial [77,78] As mentioned previously, bevacizumab with chemotherapy also showed some results in non-pancreatic NET (BETTER trial) [68]. 

One work has shown evidence of sunitinib activity in a population of 31 patients with GEP-NEN G−3, including at least six patients with NET G−3 [79]. Four out of six NET G−3 patients had partial response or stabilization of the disease [79]. In one study of 15 patients with “well-moderately” differentiated pancreatic G−3 tumors, administration of everolimus as first line treatment in four patients showed sustained disease stabilization for three out of four patients [80]. These results for both sunitinib and everolimus need to be confirmed in larger populations and are not sufficient to propose targeted therapies as first-line treatment in NET G−3.

### 4.5. Peptide Receptor Radionuclide Therapy (PRRT) 

Peptide receptor radionuclide therapy (PRRT) is a validated treatment for GEP-NET G−1 and G−2 patients with a positive SRI. 

The efficacy of PRRT (Lutetium−177 (^177^Lu)-Dotatate) and somatostatine analogues was proven in a prospective phase III study on 229 patients with advanced well differentiated midgut NET, compared with placebo and high dose of somatostine analogues [81]. The study showed a benefit in progression-free survival at 20 months with 65.2% (95% confidence interval [CI], 50.0 to 76.8) in the ^177^Lu-Dotatate group and 10.8% (95% CI, 3.5 to 23.0) in the control group. Similarly, the response rate was 18% in the PRRT group vs. 3% in the control group (*p* < 0.001) [81]. There is also available data on PRRT efficacy in pancreatic NET G−1 and G−2 [82]. Finally, a phase I study assessing the efficacy and safety of a novel SST antagonist (^177^Lu)-satoreotide tetraxetan has shown promising results in 20 patients with well differentiated NET (including one patient with NET G−3) [83].

As described above, NEN G−3 are more likely than NEC to show positive SRI uptake. There is recent data in favor of PRRT as second or third-line treatment in GEP-NEN G−3. A review of four studies with more than 10 patients with NEN G−3 treated with PRRT was performed [84]. The majority of patients had a pancreatic tumor and 50% were well differentiated. Three studies showed promising response rates (31–41%) and disease control rates (69–78%). Progression-free survival (11–16 months) and survival (22–46 months) were best for patients with a Ki−67 < 55% [84]. These results suggest that PRRT could be considered for patients with increased uptake on SRI, both in NET G−3 as well as in NEC with Ki−67 < 55%. In this sense, dual tracer using FDG-PET and SRI could provide important information for NET G−3 selection for PRRT [84,85].

### 4.6. Immunotherapy

To date, there is only little data on efficacy of immune checkpoint inhibitors in advanced NEN of the digestive tract. Nevertheless, immune checkpoint inhibitors seem a promising therapeutic option in progressive NEC or high-grade NET [86]. For instance, PD−1 blockage has shown positive results in both first and second lines treatment of Merkel cell carcinoma, a high-grade cutaneous NEC [87,88]. Some case reports have also reported tumor responses or long survival in high-grade NEN patients [86]. This could be explained by microsatellite instability, and/or high mutational load which are more pronounced in high-grade NEN. Currently, phase II studies are investigating the efficacy of immunotherapy in patients with advanced NET G−3: avelumab in metastatic or unresectable well differentiated NET G−2/G−3 (NCT03278379), avelumab in progressive NEC/NET G−3 after chemotherapy (NCT03352934), durvalumab and tremelimumab in GEP-NEN G3 (NCT03095274) and pembrolizumab in metastatic high-grade NET (NCT02939651).

## 5. Further Areas of Research

One of the main challenges regarding GEP-NET G−3 remains their initial diagnosis. Various fields of research, such as molecular biology or functional imaging, might be promising in the future to help individualize NET G−3. Indeed, whole genomic sequencing of pancreatic NET G−3 samples could help identify new structural rearrangements or mutations. Data is also needed on non-pancreatic NET G−3. The development of the NET test, a multianalyte liquid biopsy that measures NET gene expression, has shown promising results in diagnosis and predictive therapeutic assessment, and might also help individualize NET G−3. Some works have shown that radiology, such as contract-enhanced multidetector computed tomography, can help distinguish pathological grades, in particular G−1 from G−2 pancreatic NET [89]. Future radiological studies might allow to help distinguish NET G−3 from NEC or NET G−3. The second main challenge for NET G−3 lies in the choice of treatment. Here again, functional imaging could help with the use of dual tracer in order to help select patients for PRRT treatment [90]. 

## 6. Conclusions

NET G−3 represent a non-negligible entity of high-grade NEN with specific features of clinical interest. This entity represents about one third of NEN G−3 so about 5% of GEP-NEN. Prognosis and response rate seem closer to well differentiated NET G−2 than NEC, but with a worse overall survival. If diagnosis is not certain, pathologic reassessment should be readily proposed, especially in pancreatic site where NET G−3 more often occur. Treatment is not yet standardized because of lack of data and recent identification of these tumors. Available treatments include surgery, SST, liver-directed therapies and chemotherapy (mainly with alkylant-based regimen). There are interesting results with targeted therapies (sunitinib and everolimus), immune checkpoint inhibitors and PRRT that need confirmation in future studies. We propose the following algorithm (Figure 2) for therapeutic management of NET G−3 of the digestive tract. Nevertheless, treatment should be discussed in specialized meetings with NEN experts. 

## Figures and Tables

**Figure 1 jcm-09-01677-f001:**
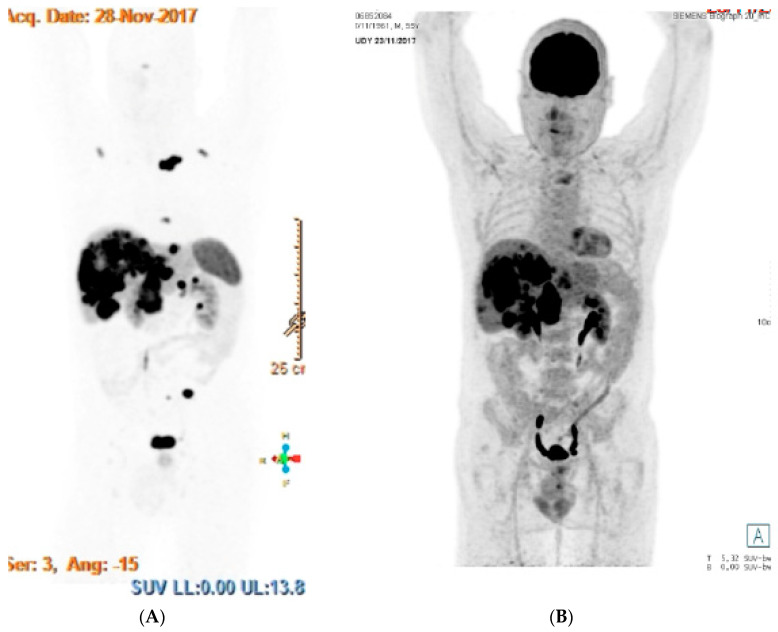
Functional imaging for a patient with metastatic pancreatic grade 3 neuroendocrine tumor (NET G−3). (**A**) Images of DOTATOC-PET on the left and (**B**) FDG-PET on the right showing both positive uptakes in the liver, lungs and nodes.

**Figure 2 jcm-09-01677-f002:**
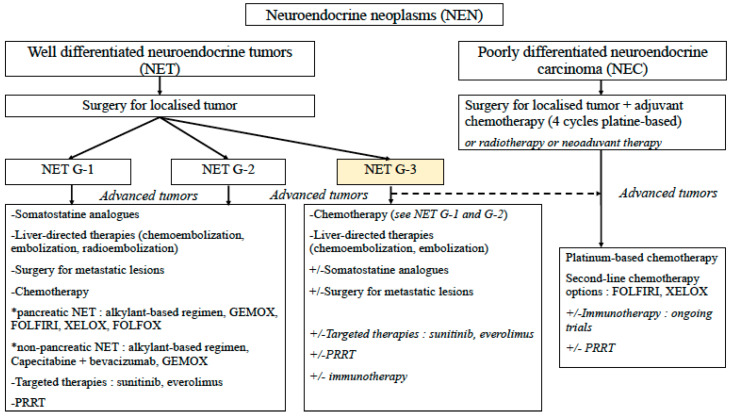
Proposed algorithm for therapeutic management of NET G−3 of the digestive tract. G−1: grade 1, G−2: grade 2, G−3: grade 3, GEMOX: gemcitabine + oxaliplatine, FOLFIRI: 5-fluorouracile + irinotecan, FOLFOX: 5-fluorouracile + oxaliplatine, XELOX: capecitabine + oxaliplatine, PRRT: Peptide receptor radionuclide therapy.

**Table 1 jcm-09-01677-t001:** The 2017 World Health Organization (WHO) Classification for Neuroendocrine Neoplasms (NEN) of the digestive tract.

Well Differentiated NEN	Ki−67 Index (%)	Mitotic Index (HPF/10HPF)
Neuroendocrine tumor (NET) G1	<3	<2/10
Neuroendocrine tumor (NET) G2	3–20	2–20/10
Neuroendocrine tumor (NET) G3	>20	>20/10
Poorly differentiated NEN		
Neuroendocrine carcinoma (NEC) G3 Small-cell type Large-cell type	>20	>20/10
Mixed Neuroendocrine-nonneuroendocrine neoplasm (MiNEN)		

NEN: neuroendocrine neoplasms, HPF: high-power fields.

**Table 2 jcm-09-01677-t002:** Main studies evaluating clinical and pathologic characteristics of patients with NET G−3.

Study, Year	Number of Patients	Median of Age (Years)	Tumor Sites	Metastatic State (%)	Median Ki−67 (%)	Median Survival (Months)
Velayoudom-Céphise et al. [3], 2013	12	56	Pancreas, non-digestive sites	100	21	41
Heetfeld et al. [12], 2015	37	52	Pancreas, rectum, stomach, small bowel	62	30	99
Basturk et al. [2], 2015	19	54	Pancreas	67	40	54
Scoazec et al. [32], 2017	21	NA	Pancreas, colorectal, stomach, small bowel	NA	35	NA
Hijioka et al. [13], 2017	21	63	Pancreas	71	29	42

NA: non-available.

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
