# Peer review of "Well Differentiated Grade 3 Neuroendocrine Tumors of the Digestive Tract: A Narrative Review"

_jcm, 2020, doi:10.3390/jcm9061677_

Round 1

Reviewer 1 Report

The present manuscript entitled ”Well Differentiated Grade 3 Neuroendocrine tumors of the digestive tract: a systematic review” by Dr Anna Pellat and et al. comments on G3 NETs.

Major comments:

The present manuscript does not fulfill the criteria of a systematic review as the

authors postulate.

It Is not clear for me why the authors present only the pancreatic cases from the Nordic study and the other types. Page 138.

The patients should only present data about G3 NETs and not about G1 and G2. For instance, at the SST, chemo and targeted therapy sections the vast majority of the text is about G1-G2 NETs. Thorough analysis is also provided about the NECs. The title is in contradiction to the test about well differentiated G3. I suggest that the authors should focus to what the title of the manuscript implies or to change title. Similar for liver directed therapies, PRRT and immunotherapy.

The algotithm lacks of evidence. Why the authors report the use of sunitinib and not of everolimus for example.

Reviewer 2 Report

Overall a well written review of the topic.

In the treatment sections there is overall great emphasis on the treatment of G1 and G2 disease and similarly in Fig 1- more emphasis should be on specifically G3 NET and NEC.

The use of additional tables to summarise the interventional trials specifically in G3 disease.

A section of further areas of research (molecular biology, genomics,  functional imaging…..) would be useful.

Line 51: HISTOPATHOLOGICAL CHARACTERISTICS OF DIGESTIVE NET G-3: Please documents any immunohistochemical differences between the NETs vs NECs if any?

Line 243: A note regarding the ECOG-ACRIN Cancer Research Group (E2211).
A randomized study of temozolomide or temozolomide and capecitabine
in patients with advanced pancreaticneuroendocrine tumors:
by Pamela L. Kunz would be useful

Line 316: Liver-directed therapies- I would remove this section if no evidence in this context

Line 351. Immunotherapy: This needs to include more recent trial reports in high grade NETs.

Reviewer 3 Report

add a section on imaging and nuclear medicine based treatment.

add images PET.

add the following references:

Clin Cancer Res. 2019 Dec 1;25(23):6939-6947. doi: 10.1158/1078-0432.CCR-19-1026. Epub 2019 Aug 22.PMID: 31439583

Nat Genet. 2018 Jul;50(7):979-989. doi: 10.1038/s41588-018-0138-4. Epub 2018 Jun 18.PMID: 29915428

Endocrinol Metab Clin North Am. 2018 Sep;47(3):485-504. doi: 10.1016/j.ecl.2018.05.002.PMID: 30098712

Clin Imaging . Sep-Oct 2018;51:59-64. doi: 10.1016/j.clinimag.2018.02.004. Epub 2018 Feb 8.

N Engl J Med. 2017 Jan 12;376(2):125-135. doi: 10.1056/NEJMoa1607427.PMID: 28076709

Neuroendocrinology. 2019;108(1):26-36. doi: 10.1159/000494258. Epub 2018 Oct 3.PMID: 30282083

Clin Radiol . 2017 Feb;72(2):150-158. doi: 10.1016/j.crad.2016.10.021. Epub 2016 Nov 24.

Reviewer 4 Report

I would suggest that the "Molecular biology" section includes some information for non-pancreatic NENs. If such info is not available, please state

The role of everolimus is not included in abstract/text/figure. This should be included and discussed.

PET-FDG --> FDG-PET

Authors refer to many of the randomised practice changing studies just to mention that data for G3NET is not available. is there any evidence useful  form retrospective analysis or small series even if outside clinical trials? many of these have been published lately and should be discussed.

Round 2

Reviewer 1 Report

The authors made a considerable revision. The review is not novel but it can consider to be published.

Author Response

Thank you for your reviewing